# Full-Length SMRT Transcriptome Sequencing and SSR Analysis of *Bactrocera dorsalis* (Hendel)

**DOI:** 10.3390/insects12100938

**Published:** 2021-10-14

**Authors:** Huili Ouyang, Xiaoyun Wang, Xialin Zheng, Wen Lu, Fengping Qin, Chao Chen

**Affiliations:** Guangxi Key Laboratory of Agric-Environment and Agric-Products Safety, College of Agriculture, Guangxi University, Nanning 530004, China; 1917392026@st.gxu.edu.cn (H.O.); zheng-xia-lin@163.com (X.Z.); luwenlwen@163.com (W.L.); Qinfengping0@163.com (F.Q.); chenao314159@163.com (C.C.)

**Keywords:** *Bactrocera dorsalis*, single-molecule real-time sequencing (SMRT), simple sequence repeat (SSR)

## Abstract

**Simple Summary:**

In this study, a full-length transcriptome was analyzed with single-molecule real-time (SMRT) sequencing, which was first used to discover simple sequence repeat (SSR) genetic markers from *B. dorsalis*. Moreover, SSR markers from isoforms were screened for the identification of species diversity. These results could provide molecular biology methods for further population research.

**Abstract:**

*Bactrocera dorsalis* (Hendel), as one of the most notorious and destructive invasive agricultural pests in the world, causes damage to over 250 different types of fruits and vegetables throughout tropical and subtropical areas. PacBio single-molecule real-time (SMRT) sequencing was used to generate the full-length transcriptome data of *B. dorsalis*. A total of 40,319,890 subreads (76.6 Gb, clean reads) were generated, including 535,241 circular consensus sequences (CCSs) and 386,916 full-length non-concatemer reads (FLNCs). Transcript cluster analysis of the FLNC reads revealed 22,780 high-quality reads (HQs). In total, 12,274 transcripts were functionally annotated based on four different databases. A total of 1978 SSR loci were distributed throughout 1714 HQ transcripts, of which 1926 were complete SSRs and 52 were complex SSRs. Among the total SSR loci, 2–3 nucleotide repeats were dominant, occupying 83.62%, of which di- and tri- nucleotide repeats were 39.38% and 44.24%, respectively. We detected 105 repeat motifs, of which AT/AT (50.19%), AC/GT (39.15%), CAA/TTG (32.46%), and ACA/TGT (10.86%) were the most common in di- and tri-nucleotide repeats. The repeat SSR motifs were 12–190 bp in length, and 1638 (88.02%) were shorter than 20 bp. According to the randomly selected microsatellite sequence, 80 pairs of primers were designed, and 174 individuals were randomly amplified by PCR using primers. The number of primers that had amplification products with clear bands and showed good polymorphism came to 41, indicating that this was a feasible way to explore SSR markers from the transcriptomic data of *B. dorsalis*. These results lay a foundation for developing highly polymorphic microsatellites for researching the functional genomics, population genetic structure, and genetic diversity of *B. dorsalis*.

## 1. Introduction

*Bactrocera dorsalis* (Hendel) (Diptera: Tephritidae), known as the oriental fruit fly, is one of the most devastating and highly invasive agricultural pests in the world, causing severe damage to over 250 species of commercial fruits and vegetables [1,2,3,4,5]. The broad range of distribution, the large number of host plants, and the complex interactions between *B. dorsalis* and diverse environments may be due to its high genetic variation, which makes it challenging to manage this pest. China’s Guangxi Zhuang Autonomous Region is an excellent vegetable- and fruit-growing region because of its superior natural geographical conditions. Moreover, Guangxi is located in the south of China, which makes *B. dorsalis* a serious pest with invasive problems and domestic outbreaks.

Simple sequence repeats (SSRs), also known as microsatellites or short tandem repeats (STRs), are tandem repeats of 1–6 nucleotides. SSR markers commonly present high levels of intra- and inter-specific variations. Because of their characteristics of assay technique, reproducibility, multi-allelic nature, codominant inheritance, abundance, and genome-wide coverage, they have been extensively applied to genetic diversity, genetic structure analysis, and paternity testing in arthropods [6,7,8,9]. In recent years, the thysanopteran insect *Frankliniella occidentalis* [10,11], coleopteran insects *Caleruca daurica* [12] and *Eucryptorrhynchus chinensis* [13], the orthopteran insect *Locusta migratoria* [14], hemipteran insects *Stephanitis nashi* and *Sclomina erinacea* [15], and dipteran insects including *Bactrocera tryoni* [16], *Glossina palpalis* [17,18], *Anopheles sinensis* [19], *Ceratitis capitata* [20], and *Aedes aegypti* [21] have been analyzed using SSRs. There have been 28,201 insect microsatellite sequences included in the National Center for Biotechnology Information (NCBI), of which 8539 belong to Order Diptera and 1670 are in Tephritidae. Existing genetic resources for *B. dorsalis* are scarce. Only 81 entries were found in the public database (NCBI 2020) under “*Bactrocera dorsalis*” as of 21 July 2020 (http://www.ncbi.nlm.nih.gov/, accessed on 1 July 2020). The known number of microsatellite markers for *B. dorsalis* population is still relatively low, which poses a barrier to explaining the genetic characteristics of the *B. dorsalis* population. Therefore, the development of more microsatellites that are suitable for *B. dorsalis* population remains to be solved. The origin of the SSR causes it to be classified as genomic-SSR (G-SSR) or expressed sequence tag-SSR (EST-SSR), which indicates whether the SSR is either in a noncoding region or a translated region [22]. In terms of SSR site development, shallow genome sequencing is more efficient than transcriptome sequencing [23], and genomic data may be mature and commonly employed in species with low polymorphism, and they can comprehensively reflect variations in a species’ DNA in coding and noncoding regions. SSR loci derived from the whole genome are generally polymorphic [24,25]; however, many species could not be sequenced at the genome level because of the high expenses. Thus, the low cost has made the development of SSRs based on RNA-seq data a mature and commonly employed method [26,27,28]. Moreover, transcriptome data may result in longer loci that are more amenable to design primers, and the primers may be more transferable to related species [23].

There is a novel third-generation sequencing technology (TGS) called the single-molecule real-time (SMRT) sequencing technology, which could generate longer reads compared to second-generation sequencing (SGS) and could meet requirements for unsolved problems in genome, transcriptome, and epigenetics research [29,30]. In addition, third-generation sequencing technology has been superior in large datasets, long sequence reads, and full-length gene transcripts, and it does not require sequence splicing and assembly at the same time. These advantages can be used in exploring new genes and discovering new SSR sites [31]. However, this technology has not previously been used in *B. dorsalis*.

A full-length transcriptomic analysis of mixed *B. dorsalis* samples at four developmental stages by using SMRT sequencing was performed. Based on the constructed transcriptome database of *B. dorsalis*, the microsatellites were explored in high throughput, aiming to lay a foundation for developing extremely polymorphic microsatellite primers and studying the functional genomics, population genetic structure, and population genetic diversity of *B. dorsalis*.

## 2. Materials and Methods

### 2.1. Transcriptome Sample Preparation

*B. dorsalis* insects were reared at the Institute of Entomology at Guangxi University (Nanning, China). Eggs were collected using a plastic cup with banana pulp and placed on moist filter paper. Larvae were subjected to a banana diet, whereas adult flies were fed with an artificial diet made from yeast extract and sugar (4:1). All of the life stages were cultured in cages at 26 ± 2 °C and 65 ± 10% humidity under a 14 h light/10 h dark photoperiod. Mixed samples of *B. dorsalis* with different insect stages were used. For further experiments, all samples were frozen in liquid nitrogen and then stored at −80 °C.

### 2.2. RNA Extraction and SMRT Sequencing

Total RNA was extracted by grinding mixed samples of *B. dorsalis* in TRIzol reagent (Life Technologies, Carlsbad, CA, USA) on ice and processed according to the protocol provided by the manufacturer. The integrity of the RNA was determined with an Agilent 2100 bioanalyzer through agarose gel electrophoresis. The purity and concentration of the RNA were determined with a Nanodrop micro-spectrophotometer (Thermo Fisher, Waltham, MA, USA). The mRNA enriched using Oligo (dT) magnetic beads was reverse-transcribed into cDNA using a Clontech SMARTer PCR cDNA synthesis kit (NO634926). The optimal amplification cycle number for downstream large-scale PCR reactions was determined through PCR cycle optimization. The optimized cycle number was used to generate double-stranded cDNA. In addition, >5 kb size selection was performed with a BluePippin^TM^ Size-Selection System, and this was mixed equally with the no-size-selection cDNA. Large-scale PCR was performed for SMRT bell library construction. cDNAs were DNA damage-repaired, end-repaired, and ligated to sequencing adapters. The SMRT bell template was annealed to a sequencing primer, bound to polymerase, and sequenced on a PacBio Sequel II platform by Gene De novo Biotechnology Co (Guangzhou, China). The raw sequencing reads of the cDNA libraries were classified and clustered into transcript consensus with the SMRT Link v5.0.1 pipeline [32] supported by Pacific Biosciences. Briefly, circular consensus sequence (CCS) reads were extracted from subread BAM files with a minimum complete pass of 1 and minimum read score of 0.65. The integrity of the transcripts was evaluated according to whether the CCS reads contained 5′ primer, 3′ primer, and poly-A structures. The sequences containing all three structures are called full-length sequences (FLs). Afterwards, primers, barcodes, and the poly-A tail trimming of complete passes were removed to produce full-length nonchimeric (FLNC) reads. Reads shorter than 50 bp were discarded. Subsequently, the entire isoform was generated by clustering the FLNC reads. A consistency sequence (rough consensus isoforms) was obtained by hierarchically clustering similar FLNC reads with minimap2. Then, the quiver algorithm was used to correct the consistency sequence. For improving the accuracy of PacBio reads, we used two strategies. First, we used non-full-length reads to polish the obtained cluster consensus isoforms with the Quiver software to produce FL, polished, high-quality consensus sequences (accuracy of ≥99%). Second, Illumina short reads were obtained from the same samples with the LoRDEC tool v0.8 [33] to further correct low-quality isoforms.

### 2.3. Functional Annotation and Structure Analysis

A reference genome (ncbi_GCF_000789215.1) sequence was used to measure the accuracy of the generated Iso-Seq reads. Then, we mapped the corrected high-quality consensus sequences to the reference genome with GMAP [34], and redundant transcripts were collapsed with a minimum identity of 95% and minimum coverage of 99%. We compared the finally obtained isoforms with the reference genome annotation and then classified them into three groups: known, novel, and new isoforms. The new isoforms were BLAST analyzed against the NCBI non-redundant protein (Nr) database (http://www.ncbi.nlm.nih.gov, accessed on 1 October 2019), Swiss-Port protein database (http://www.expasy.ch/sprot, accessed on 1 October 2019), and the Kyoto Encyclopedia of Genes and Genomes (KEGG) database (http://www.genome.jp/kegg, accessed on 1 October 2019) with the BLASTx program (http://www.ncbi.nlm.nih.gov/BLAST/, accessed on 1 October 2019) at an E-value threshold of 1e−5 for the evaluation of their sequence similarity with the genes of other species and investigation of their functions. Gene Ontology (GO) annotation was analyzed with the Blast2GO [35] software with the Nr annotation results of the isoforms. We selected the isoforms that were not shorter than 33 high-scoring segment pair hits to conduct the Blast2GO analysis and then used WEGO software to perform functional classification of isoforms [36].

Long-chain noncoding RNA (LncRNA) candidates were identified using CPC [37], a predictor of messenger RNAs based on the CNCI v2.0 [38]. LncRNA with >200 nucleotides was selected. ANGEL [39] was used to identify open reading frames (ORFs) in the full-length transcriptome of *B. dorsalis*. A previously described method was used to align all non-redundant HQ transcripts. Candidate (alternative splicing) AS events were identified using the SUPPA [40].

### 2.4. EST-SSR Detection and Primer Design

To identify microsatellites in the functional isoforms, we used the MicroSAtellite identification tool (MISA) v1.0 (http://pgrc.ipk-gatersleben.de/misa/, accessed on 1 October 2019). The SSR loci were identified based on the minimum number of repetitions of each unit size, which were 2–6, 3–5, 4–4, 5–4, and 6–4. In an interrupted composite microsatellite, the maximum number of bases for two SSRs was 100. EST-SSR primers were designed using Primer 3.0 (release 1.1.4). Mononucleotide repeats were not selected for subsequent trial analysis due to the possibility of mismatching during sequencing. The primer was designed with the following four principles. Firstly, the optimal primer length was 20 bp and could extend to 18–27 bp if needed; secondly, the optimal temperature was 60 °C and could extend to 57–63 °C; meanwhile, the difference in Tm values with upstream and downstream primers had to be less than 5 °C. Thirdly, the optimal GC content was 50% and could extend to 30–70%; fourthly, the PCR amplicon should have had a length of 100–300 bp [41].

### 2.5. Amplification and Validation of EST-SSRs

In the primer polymorphism verification experiments, nondenaturing polyacrylamide gel was used. In order to screen out primers with polymorphisms, the 80 primer pairs mentioned above were synthesized by the Shanghai Biological Engineering (Shanghai) Company. Polymerase chain reaction (PCR) was performed in a 10 μL reaction volume containing 5 µL of 2×Taq PCR MasterMix (TIANGEN Co., Ltd. Shanghai, China), configuring the PCR system according to the reagent instructions. The PCR system also included 1 μL of template DNA (Table 1), 0.5 μL of each primer (10 μmol/L), and 3 μL of ddH_2_O. PCR amplification was performed using the following temperature program in PCR Amplifier (T100^TM^ Thermal Cycler, BIO-RAD, Hercules, CA, USA): melting at 94 °C for 3 min; 32 cycles of denaturation at 94 °C for 30 s, with an annealing temperature depending on the primer for 30 s, and extension at 72 °C for 30 s; extension at 72 °C for 5 min; preservation at 4 °C. We used 8% nondenaturing polyacrylamide gel to run the PCR amplification products on a vertical plate electrophoresis apparatus when selecting polymorphic primers.

## 3. Results

### 3.1. SMRT Sequencing

The PacBio Sequel system generated a total of 40,319,890 subreads with an average read length of 1009 bp. The N50 was 1659 bp. After self-correction and merging, 535,241 circular consensus sequences (CCSs) (Figure 1A) with a mean length of 1756 bp were formed. The subreads also generated 386,916 full-length non-chimeric sequences (FLNCs) (Figure 1B) with an average length of 1622 bp. A total of 22,780 high-quality (HQ) isoforms with ≥99% accuracy were obtained after clustering and removing redundant sequences (Figure 1C). These isoforms were identified and mapped to 12,274, corresponding to 5365 known isoforms and 5321 new isoforms that were not annotated previously (Table 2).

### 3.2. Functional Annotation

The functions of the HQ transcripts were annotated; then, the clusters of the Nr, Swiss-Port, GO, and KEGG annotations were used to elucidate the functions of the non-redundant isoforms. Of the 12,274 transcripts analyzed, 1124 could not be functionally annotated to any of the used databases, and 2412 were shared among the four databases (Figure 2). After that, all transcripts were aligned to the NCBI non-redundant protein database (Nr). With its comprehensive content and the inclusion of species information in the annotated results, the Nr database is a non-redundant protein database that can be used to classify homologous species. The results showed that 11,150 (90.84%) HQ transcripts were annotated using Nr and exhibited homology with known proteins of various species, including *B. dorsalis* (4663, 41.82%), *B. cucurbitae* (178, 1.60%), and *C. capitata* (164, 1.47%) (Table 3). Moreover, *Bactrocera* was the dominant genus by far, accounting for approximately half of all Nr annotation results. In total, 6102 (49.71%) isoforms were annotated into 49 sub-categories of the three main GO categories: biological processes (BPs), cellular components (CCs), and molecular functions (MFs) (Figure 3). Among them, 20 sub-categories, 18 sub-categories, and 11 sub-categories were annotated in BPs, CCs, and MFs. The top ten sub-categories were cellular processes (3240), metabolic processes (3138), binding (2972), catalytic activity (2962), single-organism processes (2961), cells (2472), cell parts (2472), organelles (1798), macromolecular complexes (1251), and biological regulation (1118). In addition, 3169 (25.82%) isoforms were identified in the Kyoto Encyclopedia of Genes and Genomes (KEGG) database and grouped into 162 KEGG pathways that were categorized into five broad categories: metabolism (1832, 57.81%), cellular processes (364, 11.49%), organismal systems (173, 5.46%), environmental information processing (276, 8.71%), and genetic information processing (1039, 32.79%) (Table 4). Among all of the sub-categories, ‘metabolic pathways’, ‘biosynthesis of secondary metabolites’, and ‘oxidative phosphorylation’ were the top three. Using Swiss-Port, 8,675 isoforms were annotated. Then, the CPC (http://cpc.cbi.pku.edu.cn/, accessed on 1 October 2019) and CNCI were used to predict the coding ability of the new genes and transcripts.

### 3.3. Predictive Analysis of SSRs

Among the 12,274 evaluated sequences, 1978 SSRs were identified from the 1714 SSR-containing sequences. The frequency of occurrence of SSRs was 13.96% (total number of HQ transcripts containing SSRs/total number of HQ transcripts examined). The distance of the average distribution was 11.00 kb (total length/total number of independent genes used to find SSRs), and the SSRs appeared at a frequency of 16.12% (total number of SSRs identified/total number of HQ transcripts examined). The number of repetitive SSR motifs was 105. The identification results showed that there were a total of 286 transcript sequences with more than one EST-SSR locus, and a total of 52 SSRs were presented in compound form. Mononucleotide repeats are prone to mismatch during sequencing, leading to low sequencing quality, so they could not be selected in subsequent experimental analyses.

Among all of the SSR repeat types, AT/AT dinucleotide repeats repeated the most, accounting for 19.77% of all repeat motifs, followed by AAC/GTT (17.29%) and AC/GT (15.42%) (Figure 4). Furthermore, trinucleotide repeats contained the most repeat types—up to 38—and the main repeat types consisted of AAC/GTT (17.29%), AGC/CTG (6.72%), and ACC/GGT (5.26%). The following tetranucleotide repeats also included 35 repeat types. As shown in Table 5, the highest number of identified SSRs were trinucleotide repeats (875, 44.24%), followed by dinucleotide repeats (779, 39.38%) and tetranucleotide repeats (302, 15.27%). The rarest type of EST-SSR was the pentanucleotide type (0.46%), but not the hexanucleotide type (0.66%).

Among all of the SSR repeat motif lengths, the lengths ranged from 12 to 190 bp. A total of 1638 SSRs were mainly concentrated in the 12–18 bp region (88.02%), 109 SSRs were 20–26 bp long (5.86%), 16 SSRs were 27–33 bp long (0.86%), and 98 SSRs were longer than 33 bp (5.27%) (Figure 5).

As for SSR tandem repeats, the most common tandem repeat number was 5 (735, 37.16%), followed by 6 (490, 24.77%) and 4 (288, 14.56%). Furthermore, the motif repeats of 1952 SSRs were less than 9, accounting for 98.69% of all identified SSR loci (Figure 6).

### 3.4. Verification of Novel and Polymorphic EST-SSRs

The development of primers contributes to the basis for further research on the genetic structure and diversity of species. In our study, 174 samples from different geographical populations of Guangxi were subjected to PCR amplification using 80 pairs of newly developed EST-SSRs. Of these 80 EST-SSRs, although five failed to generate a product, the rest of the 75 primer pairs successfully resulted in amplification. Among the 75 primer pairs, six exhibited poor universal applicability, three produced multiple bands, and 12 were monomorphic. Of the remaining 54 primer pairs with the capacity to generate polymorphic amplification products, 13 primer pairs generated unstable and unclear amplification, and the 41 others produced stable and clear amplification products. Details of these 41 primer pairs are shown in Table 6. The detailed information of the electrophoretogram of the nondenaturing polyacrylamide gel is shown in Appendix A and Appendix A.

## 4. Discussion

The transcriptome data were compared with the reference genome of *B. dorsalis*. The percentage of the sequences in the reference genome was more than 76%. More than 75% of the sequences were unique-mapped against the reference genome, and no more than 2% were multiple-mapped. In addition, 90.84% of the query sequences had comments for the four databases, indicating that the quality of the transcriptome was satisfactory.

In the study of gene annotation, to obtain gene function information, we can classify lots of new transcripts. Of the Nr databases, 4663 were annotated in *B. dorsalis*, accounting for 41.82% of the total annotated isoforms. This result not only indicates the close relationship between *B. dorsalis* and *Bactrocera* species, but also explains the correct assembly and annotation of this transcript library. A metabolic pathway analysis of 3169 isoforms of *B. dorsalis* was carried out with the KEGG database, which provided essential data for the next step of the mining of functional genes and functional verification of target genes.

The content of microsatellite sequences and dominant microsatellite sequence types varied among different species. As shown in Table 7, the frequency of occurrence of SSR sites showed differences among species. In the present study, the frequency was higher than that in *Bemisia tabaci* (5.07%), *Conopomorpha sinensis* (15.25%) [42], *Odontotermes formosanus* (9.98%), *Galeruca daurica* (5.36%) [12], *Tenebrio molitor* (1.67%), *Arma chinensis* (7.60%), *Tomicus yunnanensis* (1.29%) [43], *Eucryptorrhynchus chinensis* (10.36%) [13], *Bicyclus anynana* (3.15%) [44], *Plodia interpunctell* (8.52%) [45], *Rhyacionia leptotubula* (3.09%) [46], *Plutella xylostella* (6.59%), *Conopomorpha sinensis* (15.25%) [42], and *Athetis lepigone* (2.96%) [47]. We deduced that the investigation of SSR sites might relate to the database size, search element screening SSR site sequence criteria, RNA quality, and species specificity [48]. The number and density of SSR sites identified in this study were significantly higher than the results of the EST development of *B. dorsalis* in 2014, which was possibly due to the different sequencing depth and assembly quality [49]. The largest abundance of trinucleotide repeat motifs is consistent with that of other insect species, such as *Blattella germanica* [50], *Laodelphax striatellus* [51], *Tetranychus urticae* [52], *Anopheles sinensis* [19], *Dolerus aeneus* [53], *Tomicus yunnanensis* [43], *Nilaparvata lugens* [54,55], *Mythimna separata,* and *Anoplophora chinensis* [48]. This phenomenon is most likely since trinucleotide may be more stable than other repeat unit types in the protein-coding region and rarely produces sliding mutations of the coding frame. Moreover, *Ixodes scapularis* [52], *Grapholitha molesta* [56], *Leptinotarsa decemlineata* [57], *Sitodiplosis mosellana* [58], *Tenebrio molitor*, *Cimex lectularius*, and *Phenacoccus solenopsis* [59] are dominated by single nucleotide repetitions; *Drosophila melanogaster*, *Epacromius coerulipes* [60], *Pardosa pseudoannulata* [61], *Gampsocleis gratiosa* [62], and *Apis mellifera* have the most abundant repeat types with dinucleotides. *Bombyx mori* [63] is dominated by tetranucleotides. *Tribolium castaneum* and bee (*Apis*) [64] were mainly based on hexanucleotides, indicating that the differences in SSR abundance, base composition, and dominant base type among species are not significantly correlated with species kinship. Contrary to the study of Lepidoptera insects, such as *Athetis lepigone* and *Mythimna separata* [65], GC/CG is rare in the dinucleotides in animal or plant transcriptomes or genomes. In addition, the present study confirms that trinucleotide repeats are the most abundant in transcribed regions. The reason for such polymorphisms is still under debate, although it seems to be slippage events during DNA replication.

## 5. Conclusions

In the current study, we obtained 76.6 Gb of clean reads, and approximately 99.3% of the high-quality transcripts were almost greater than 1000 bp in length. In addition, 12,274 sequences were successfully annotated, and 1978 EST-SSRs—excluding mononucleotide repeats—were identified. Moreover, because of their high polymorphism, 41 of these EST-SSRs were proved to be reliable molecular markers in research on *B. dorsalis*. Overall, with these large amounts of transcription data, most genetic analyses, such as the discovery of new genes, gene function verification, and study of genetic diversity, will be facilitated. In addition, the markers mentioned above will help reveal the genetic relationships of *B. dorsalis* and its related species in terms of functional molecular markers.

## Figures and Tables

**Figure 1 insects-12-00938-f001:**
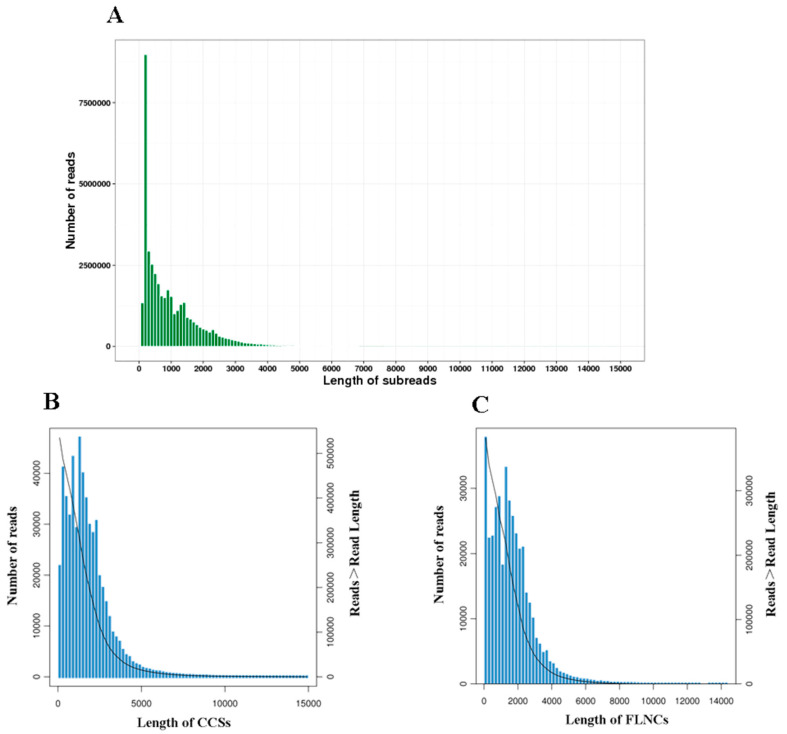
SMRT sequencing of *B. dorsalis* transcriptomes. (**A**) Length distribution of subreads. (**B**) Length distribution of CCSs. (**C**) Length distribution of FLNCs. Note: The left ordinate represents the number of sequences of this length, and the right ordinate represents the number of sequences whose length is greater than a certain value (X-axis).

**Figure 2 insects-12-00938-f002:**
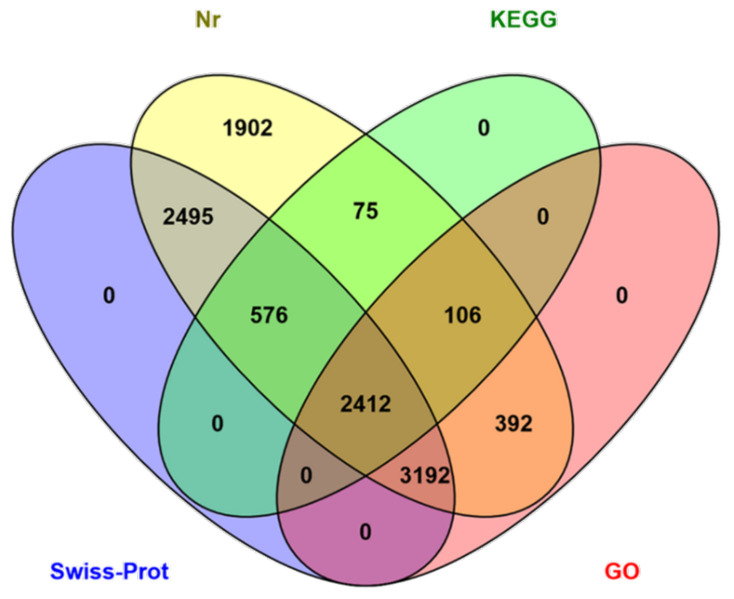
Venn diagram of the number of the annotation results from the four databases.

**Figure 3 insects-12-00938-f003:**
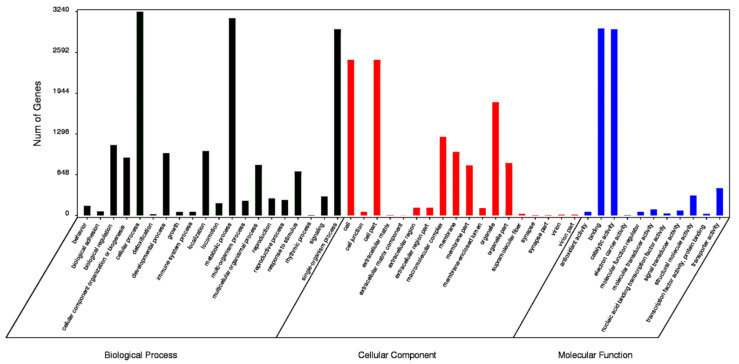
Gene Ontology (GO) distribution for *B. dorsalis* isoforms.

**Figure 4 insects-12-00938-f004:**
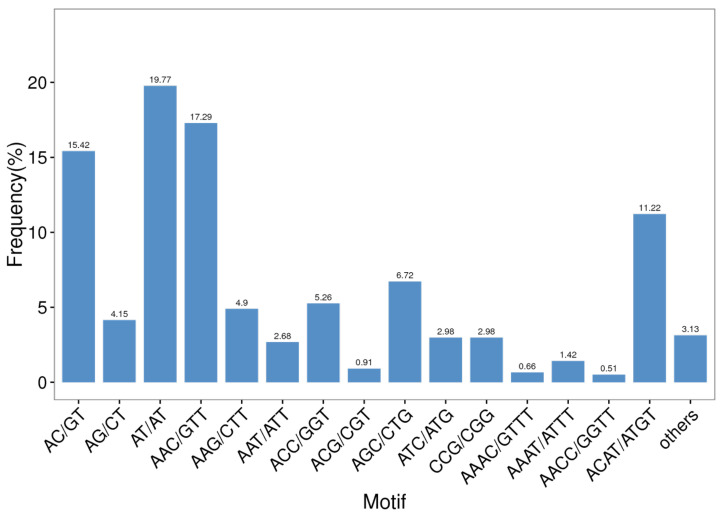
Microsatellite distribution on different repeat motifs (considering sequence complementary) in the *B. dorsalis* transcriptome.

**Figure 5 insects-12-00938-f005:**
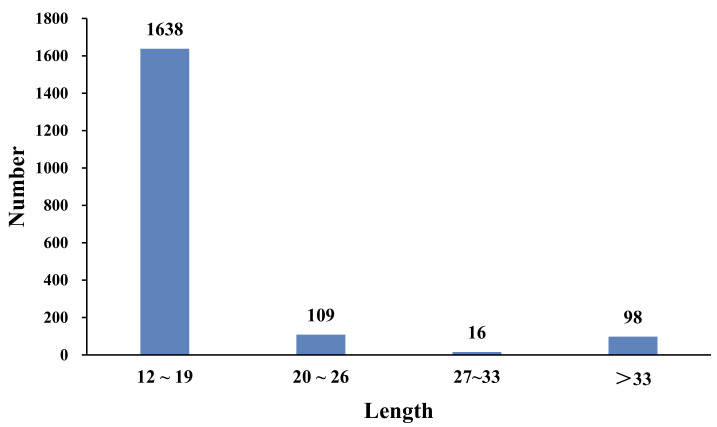
Length distribution of microsatellites in the *B. dorsalis* transcriptome.

**Figure 6 insects-12-00938-f006:**
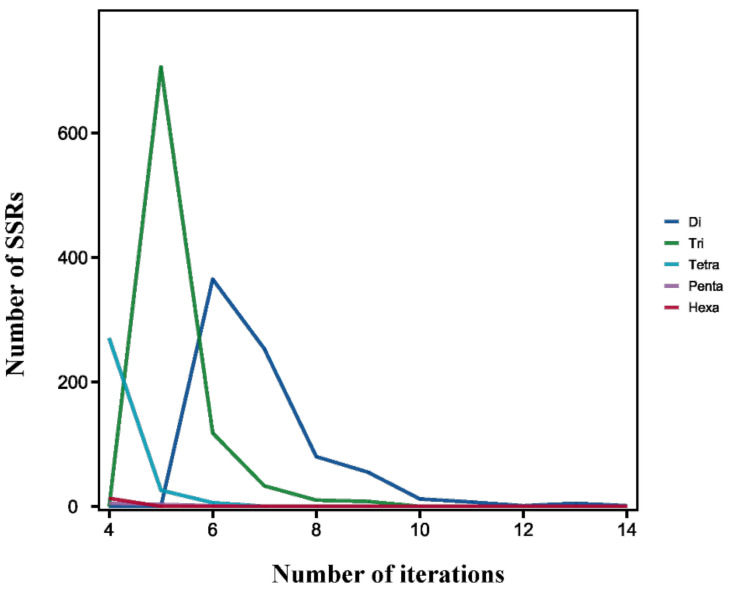
The distribution of different types of repeat units of SSRs in *B. dorsalis*.

**Table 1 insects-12-00938-t001:** The *Bactrocera dorsalis* populations used in this study.

Collection Location		Population Code	Number	Host Plants	Geo-Coordinates	Collection Date
City/Province	Town					
Beihai	Tieshangang	BHTSG	8	Guava	N21°31′45″ E109°25′18″	2020.10.24
	Hepu	HPSK	8	Citrus Reiculata Blanco	N21°44′36″ E109°22′52″	2020.10.24
	Weizhou island	WZD	8	Pawpaw	N21°01′39″ E109°05′24″	2021.6.9
Yulin	Yuzhouqu	YLYZQ	8	Guava	N22°41′35″ E110°07′43″	2020.10.29
	Luchuan	YLLC	8	Guava	N22°9′39″ E110°14′33″	2020.11.1
	Rongxian	RXRX	8	Citrus Reiculata Blanco	N22°48′33″ E110°28′47″	2020.11.25
Chongzuo	Longzhou	CZLZ	8	Green date, Tangerine	N25°16′16″ E110°19′51″	2020.3.31
	Jiangzhouqu	CZJZQ	6	Citrus Reiculata Blanco	N22°39′34″ E107°38′21″	2020.10.31
Baise	Tianyang	BSTY	6	Mango	N23°44′08″ E106°54′56″	2020.10.31
Guigang	Gangnanqu	GGGNQ	8	CitrusGonggan, sugar orange	N23°01′55″ E109°49′41″	2020.10.25
	Pingnan	GGPN	8	Citrus Reiculata Blanco	N23°25′17″ E110°32′11″	2020.11.25
Guilin	Yongfu	GLYF	8	Momordica Grosvenori	N24°59′31″ E109°59′54″	2020.11.1
	Gongcheng	GLGC	8	Gongcheng Persimmon	N24°43′41″ E110°52′31″	2020.11.2
Wuzhou	Mengshan	WZMS	8	Sugar Orange	N24°11′38″ E110°31′29″	2020.10.28
Fangchenggang	Shangsi	FCGSS	8	Guava	N22°1′33″ E108°0′3″	2020.9.17
Nanning	Longan	NNLA	8	Mango	N23°9′57″ E107°41′46″	2020.7.17
Hechi	Duan	HCDA	6	Citrus Reiculata Blanco	N23°58′45″ E108°5′57″	2020.11.16
Laibin	Xincheng	LBXC	4	Citrus Reiculata Blanco	N23°50′54″ E108°49′54″	2020.11.1
Qinzhou	Qinnanqu	QZQN	8	Guava	N21°56′19″ E108°39′25″	2021.7.26
Guangdong, Guangzhou		GZ	8	Uncertain	N23°9′29″ E113°21′28″	2020.10.17
Luoyang, Henan		HN	8	Peach	N34°39′55″ E112°24′55″	2021. 5.1
Baoshan, Yunnan		YN	6	Persimmon	N25°7′16″ E99°9′57″	2020.2.5

**Table 2 insects-12-00938-t002:** Data filtering results of the *B. dorsalis* transcriptome.

Categories	Sequencing	Amount
Subreads	Number of subreads	40,319,890
	Number of bases	40,685,843,094
	Average length	1009
	N50	1659
Circular consensus sequence (CCS)	Number of sequences	535,241
	Number of bases	940,115,109
	Mean length	1756
Full-length non-concatemer (FLNC)	Number of sequences	386,916
	Mean length	1622
High-quality isoforms	Number of sequences	22,780 (99.30%)
	Mean length	1783
	N50	2256
Comparison of reference genomes	Total mapped	17,459 (76.64%)
	Multiple mapped	362 (1.59%)
	Unique mapped	17,097 (75.05%)
Gene expression	All mapped genes	4912
	All mapped isoforms	12,274
	Known isoforms	5365 (43.71%)
	Novel isoforms	226 (1.84%)
	New isoforms	6683 (54.45%)

**Table 3 insects-12-00938-t003:** The species distribution of the Nr annotation results.

Genus	Name of Species	Number of Isoforms	Percentage (%)	
			Individual	Generic Species
*Bactrocera*	*B. dorsalis*	4663	41.82	44.88
	*B. cucurbitae*	178	1.60	
	*B. latifrons*	98	0.88	
	*B. oleae*	65	0.58	
*Ceratitis*	*C. capitata*	164	1.47	1.47
*Drosophila*	*D. melanogaster*	87	0.78	2.46
	*D. grimshawi*	71	0.64	
	*D. persimilis*	58	0.52	
	*D. sechellia*	43	0.39	
	*D. busckii*	15	0.13	
*Rhagoletis*	*R. zephyria*	26	0.23	0.23

**Table 4 insects-12-00938-t004:** KEGG pathway mapping for *B. dorsalis*.

KEGG Category	Sub-Pathways	Genes
Metabolism (57.81%)		
	Metabolic pathways	869
	Biosynthesis of secondary metabolites	367
	Oxidative phosphorylation	330
	Biosynthesis of antibiotics	317
	Microbial metabolism in diverse environments	309
	Carbon metabolism	278
	Glycolysis/Gluconeogenesis	246
	Biosynthesis of amino acids	200
Cellular processes (11.49%)		
	Phagosome	159
	Lysosome	107
	Endocytosis	85
	Peroxisome	46
	Focal adhesion	17
	Adherens junction	17
	Tight junction	17
	Regulation of actin cytoskeleton	17
	Quorum sensing	8
	Regulation of autophagy	5
	Gap junction	3
Organismal systems (5.46%)		
	Phototransduction—fly	82
	Longevity regulating pathway—multiple species	47
	Dorso-ventral axis formation	19
	Platelet activation	14
	Leukocyte transendothelial migration	14
	Thyroid hormone signaling pathway	14
	Oxytocin signaling pathway	14
Environmental information processing (8.71%)		
	ECM–receptor interaction	61
	Hippo signaling pathway—fly	54
	Wnt signaling pathway	30
	Two-component system	28
	FoxO signaling pathway	22
	ABC transporters	17
	Phosphatidylinositol signaling system	16
	TGF-beta signaling pathway	16
	Rap1 signaling pathway	14
	Hippo signaling pathway	14
Genetic information processing (32.79%)		
	Ribosome	280
	RNA transport	170
	Protein processing in endoplasmic reticulum	151
	RNA degradation	92
	Proteasome	76
	Spliceosome	74
	Aminoacyl-tRNA biosynthesis	62
	Ubiquitin mediated proteolysis	54
	Protein export	47
	mRNA surveillance pathway	40

**Table 5 insects-12-00938-t005:** Amount and distribution of SSR sites in the full-length transcriptome of *B. dorsalis*.

Repeat Type	Repeat Number								Total	Percentage (%)
	4	5	6	7	8	9	10	11~14		
Dinucleotide			365	253	80	55	12	14	779	39.38
Trinucleotide		706	118	33	10	8			875	44.24
Tetranucleotide	270	26	6						302	15.27
Pentanucleotide	5	3	1						9	0.46
Hexanucleotide	13								13	0.66
Total	288	735	490	286	90	63	12	14	1978	100.00
Percentage (%)	14.56	37.16	24.77	14.46	4.55	3.19	0.61	0.71		

**Table 6 insects-12-00938-t006:** Primer sequences for microsatellite markers and optimal PCR conditions.

Name	Motif Type	Primer Sequence	Tm (°C)	Size (bp)
BdSSR1 (Isoform 2924)	(TACA)4	GGCAACCAATAGAACTGGGA	53	280
		GTGCAAAAGTGTGTGCGTTT		
BdSSR2 (Isoform 3402)	(AAAC)4	CGCGAATACTACGGACTTTAGG	53	278
		CAACCTACCCACATCTACACACA		
BdSSR3 (Isoform 3866)	(TATG)4	ATATCACCGCCGTAGCAAAC	52	279
		TTGGCGTCAATCATAGCGTA		
BdSSR4 (Isoform 4371)	(ACAT)4	TGCCATATGGTTGCATCAGT	49	180
		GAAGCGCGAATGAACAAAAT		
BdSSR5 (Isoform 4367)	(CAG)5	AAAGTAAATGTTGCGGTCGG	51	258
		GTATAGCGCCGGTGATGAGT		
BdSSR6 (Isoform4777)	(ATAC)4	AGCCCAGAAACTCACAGCAT	50	197
		AACCGCAACAAAACAATTCC		
BdSSR7 (Isoform 4926)	(TA)7	CGATAGCGCCCTATTTGTGT	48	144
		CATTTGCGGTGCATTATTTG		
BdSSR8 (Isoform 5257)	(TAG)5	TGTGACGGGTTGCTACCATA	49	166
		CGCAAAAACAAGACCCAAAT		
BdSSR9 (Isoform 84)	(GGC6)	GCGACAAACAGTGCTTACGA	53	233
		CCGCTGCTGTAAGAGGACTT		
BdSSR10 (Isoform 1104)	(AAC)6	GCTTGTTGTTGTTGTGGTGG	53	238
		ACGAAACGAGTGCGAAGAGT		
BdSSR11 (Isoform 1807)	(ATAC)5	TTGAAACGCGTTGAAAAGTG	50	269
		CGTTGCACTCAGGACTACGA		
BdSSR12 (Isoform 2327)	(TACA)4	CATCGGGAAGTGCCAGTTAT	52	264
		TGCCCAACATGTTATCTGGA		
BdSSR13 (Isoform 2402)	(ATTT)4	GCTGGCCTACTCAGCGTATC	53	217
		CTGCCCCGGTTAAAGTACAA		
BdSSR14 (Isoform 2859)	(ACAT)4	GCGAAAGCGTAAAGGTGTGT	47	132
		TTCAAAGTTAATGCGAAGCA		
BdSSR15 (Isoform2922)	(TACA)4	GTGCAAAAGTGTGTGCGTTT	51	155
		TCATCGGCCAATTCGAGTAT		
BdSSR16 (Isoform 2923)	(TACA)4	GTGCAAAAGTGTGTGCGTTT	51	155
		TCATCGGCCAATTCGAGTAT		
BdSSR17 (Isoform 2923)	(CAG)5	GCAAGAAAAGCAGCAAAACC	51	170
		GCTCGGCGAGTAACTCATTC		
BdSSR18 (Isoform 3194)	(CAAGAG)4	GGCCAAACAGAATGAGGAAA	51	200
		GCTACTACGCTTTCTTGCGG		
BdSSR19 (Isoform 3393)	(AGC)5	CAATAGTGCGAGCAGTCGAA	51	173
		GCAACGTTTCGTGATTCTCA		
BdSSR20 (Isoform 3695)	(GCTCCA)4	TATACGGCTCCCTACATCGC	54	181
		CACTTGGTGCAACCAGCTTA		
BdSSR21 (Isoform 3954)	(ACAT)4	ACACACGAAGCGGAAGAGTT	53	278
		CTGCCTCTCGTGTTTGCTTA		
BdSSR22 (Isoform 4461)	(GCT)6	GTAATTGTGCCGTTCGAGGT	53	217
		CCGGACTGCTATCCACATTT		
BdSSR23 (Isoform 4478)	(AAC)5	GTCAGCTCTGGAGTCGGAAC	55	249
		GGTGGTGTCTGTTGTCGTTG		
BdSSR24 (Isoform 4488)	(ATAC)5	AGCAGCTGAAGAGGAAGTGC	53	250
		TATGTAGAAACGGTTCGGGC		
BdSSR25 (Isoform 4513)	(ATAC)4	GCGAAGCGGACAAAAGTTAG	52	192
		TTTCTGCACTTCGCACTATCA		
BdSSR26 (Isoform 4606)	(CAG)5	CAGCGAACAGGAGCACATTA	50	236
		CGTATTGCATCATTTGTGGC		
BdSSR27 (Isoform 4611)	(CAG)5	CAGCGAACAGGAGCACATTA	50	236
		CGTATTGCATCATTTGTGGC		
BdSSR28 (Isoform 4614)	(CAG)5	CAGCGAACAGGAGCACATTA	50	236
		CGTATTGCATCATTTGTGGC		
BdSSR29 (Isoform 4621)	(AT)7	TGTATGTACGCACACCAGCA	51	114
		AACACAAATGCGGCTTCTTT		
BdSSR30 (Isoform 4621)	(CAG)5	CAGCGAACAGGAGCACATTA	50	236
		CGTATTGCATCATTTGTGGC		
BdSSR31 (Isoform 4654)	(TATG)4	AGTTTTCGCTGCCGCTATTA	52	214
		CGGCCATCTCGTAGGTATGT		
BdSSR32 (Isoform 4707)	(AC)6	GCTAGTTTGACGATGAGGGC	53	174
		CAGCACGTAATTTGCTGCAC		
BdSSR33 (Isoform 4731)	(ACAT)4	TCCAACAGCAAATTCGACAA	48	234
		TCTCATAAAAGCGCATACAAAAA		
BdSSR34 (Isoform 4932)	(CATA)4	CAACGCTCACTCGCTCATTA	49	190
		AATGTTCCGAATTTTCGTCG		
BdSSR35 (Isoform 4950)	(CAA)5	GGTGCTGGTGGCAGTTTATT	54	142
		TTGTTGTAGCGGTGGTGGTA		
BdSSR36 (Isoform 4980)	(CA)7	TCCATGAGATCGAATGCAAA	49	278
		CGATTCTAACTGCGAACGAA		
BdSSR37 (Isoform 4992)	(CAA) 5	ACTCGCATTGAATGGACACA	52	174
		AAATGATGCTGCTGCTGATG		
BdSSR38 (Isoform 5040)	(ATAC) 4	GGATACTAGTGGTGGTCCGC	54	175
		GCAGCTAGGATGCACAACAA		
BdSSR39 (Isoform 5126)	(TACA) 4	ACAGCCGAGTTTGAGCTTGT	50	245
		TTGCATGAAAAGCAAACACC		
BdSSR40 (Isoform 5749)	(GCT)5	AAGACGAAGAAGATGCGGAA	51	157
		AAGACGAAGAAGATGCGGAA		
BdSSR41 (Isoform 5805)	(AGC)5	ACAGCAACAACAGCAACAGC	55	225
		TGTGTGCTAGAAGACGCACC		

**Table 7 insects-12-00938-t007:** Dominant repeat types in different insects (incomplete statistics).

Dominant Repeat Type	Order	Species	Frequency (%)	Omics Level	Reference
Mononucleotide	Hemiptera	*Cimex lectularius*	18.68	Transcriptome	Li et al., 2019
		*Stephanitis nashi*	26.87	Transcriptome	Xie et al., 2019
	Lepidoptera	*Conopomorpha sinensis*	15.25	Transcriptome	Meng et al., 2017
		*Mythimna separata*	11.51	Transcriptome	Li et al., 2017
		*Grapholitha molesta*	13.16	Transcriptome	Leng et al., 2018
	Thysanoptera	*Frankliniella occidentalis*	18.95	Transcriptome	Duan et al., 2012
	Coleoptera	*Galeruca daurica*	5.36	Transcriptome	Zhang et al., 2016
		*Tenebrio molitor*	1.67	NA	Zhu et al., 2013
		*Tribolium castaneum* *	10.87	Transcriptome	Zhang et al., 2008
		*Leptinotarsa decemlineata*	NA	Genome	Liu et al., 2018
	Diptera	*Sitodiplosis mosellana*	13.47	Transcriptome	Duan et al., 2011
		*Anopheles sinensis*	NA	Genome	Wang et al., 2016
	Homoptera	*Phenacoccus solenopsis*	6.33	Genome	Luo et al., 2014
Dinucleotide	Lepidoptera	*Plodia interpunctell*	8.25	Transcriptome	Tang et al., 2017
		*Rhyacionia leptotubula*	3.09	Transcriptome	Zhu et al., 2013
	Hemiptera	*Arma chinensis*	7.6	NA	Li et al., 2019
	Orthoptera	*Epacromius coerulipes*	44. 17	Transcriptome	Jin et al., 2015
		*Gampsocleis gratiosa* *	18.64	Transcriptome	Zhou et al., 2019
	Hymenoptera	*Apis mellifera*	10.804	Genome	Zhao et al., 2007
Trinucleotide	Coleoptera	*Tomicus yunnanensis*	1.29	Transcriptome	Yuan et al., 2014
		*Anoplophora chinensis*	25.31	Transcriptome	Han et al., 2019
		*Eucryptorrhynchus chinensis*	10.36	Transcriptome	Wu et al., 2016
	Lepidoptera	*Dolerus aeneus*	NA	NA	Cook et al., 2011
		*Athetis lepigone*	2.96	NA	Li et al., 2013
		*Mythimna separata*	1.93	Transcriptome	Hu et al., 2015
		*Plutella xylostella*	6.59	Transcriptome	Ke et al., 2013
	Hemiptera	*Nilaparvata lugens*	NA	Transcriptome	Liu et al., 2010
		*Sclomina erinacea*	5.67	NA	Li et al.,2019
	Homoptera	*Laodelphax striatellus*	NA	Transcriptome	Zhang et al., 2010
	Orthoptera	*Gampsocleis gratiosa* *	39.38	Genome	Zhou et al., 2019
	Diptera	*Anopheles sinensis* (2014)	NA	Transcriptome	Zhou et al., 2018
		*Bactrocera dorsalis*	NA	Transcriptome	Wei et al., 2014
	Blattaria	*Blattella germanica*	NA	Genome	Wang et al.,2015
Tetranucleotide	Lepidoptera	*Bombyx mori*	26.51	Transcriptome	Mi et al., 2011
Pentanucleotide	NA	NA	NA	NA	
Hexanucleotide	Coleoptera	*Tribolium castaneum **	13.65	Genome	Zhang et al., 2008
	Hymenoptera	Bee (*Apis*)	10.52	Transcriptome	Li et al., 2004

Note: The * represents the frequency at different levels of omics of the same species. The NA means not available.

## Data Availability

The submission of raw sequences to NCBI SRA has been started undern project number PRJNA695387. Data are contained within the article or Appendix A.

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
