# Peer review of "Full-Length SMRT Transcriptome Sequencing and SSR Analysis of Bactrocera dorsalis (Hendel)"

_insects, 2021, doi:10.3390/insects12100938_

Round 1

Reviewer 1 Report

In the manuscript “Full-length SMRT transcriptome sequencing and SSR analysis of Bactrocera dorsalis (Hendel)” by  Ouyang et al. authors analyzed Bactocera dorsalis transcriptome in details. It provides information that can be used by the researcher for further research.  I recommend this manuscript for publication after the suggested corrections.

I suggest that the manuscript to be read by an English speaker as English needs editing.  

Line 10: Change “firstly” to “first”

Line 11: Change “solutions” to “methods”

Line 14: Change “has caused” to “causes”

Line 15: Delete the word “aiming”

Line 38: Delete the word “its”

Line 39: Remove the word “the” before B. dorsalis

Line 42: Change the word “planting” to “growing”

Line 42: Rephrase “Moreover, Guangxi locates in the southern tip of China…………………….”

Line 50-53: Use lower case first letter for “Thysanopteran” , “Coleopteran” , “Orthopteran”, “Hemipteran”, “Dipteran”

Line 51-53: Use lower case alphabets for species name.

Line 56: Change the word “are” to “belong to” before Order.

Line 56: Uncapitalize the word “Family”

Line 59: Use the word “known” before number

Line 60: Change “a more profound explanation of the” to “explain”

Line 61: Rephrase “Therefore, how to develop more microsatellites suitable for the B. dorsalis population remains to be solved.

Line 71: Replace “mature and commonly employed” with “preferable”

Line 75: Remove the word “provide”

Line 76: Replace the word “with” to “to”. Delete the word “one”

Line 76: Rephrase “making it meet requirements for unsolved problems in genome, transcriptome, and epigenetics research”

Line 81: Delete “according to existing second-generation transcriptional data”

Line 88: Change the word “the study of” to “studying”

Line 98: Change the word “states” to “stages”

Line 98: What do you mean by test worm?

Line 103: Do you mean ice or was “dry ice” used?

Line 103: Change “by the” to “according to”

Line 135: Rephrase “A reference genome (ncbi_GCF_000789215.1) sequence assembly was against to measure the accuracy of the generated Iso-Seq reads.”

Line 180: Change “predegeneration: to “melting”

Line 227: Uncapitalize “Oxidative”

Line 250: Uncapitalize “Alternative”

Line 251: Uncapitalize “Alternative”, “Mutually”, “Retained”, “Alternative”

Line 256: Delete “the abovementioned”

Line 313: Rephrase “The single ratio of the sequence sequencing in the reference genome to the total was more than …………………”

Line 316: What do you mean by comments?

Line 348: Provide scientific name of the Bee

Line 349: Is it single nucleotide or tetranucleotide

Line 350: Provide scientific name of bees

Line 355: Delete “this conclusion”

Tables and Figures

Table 1: Provie more details of location, are the name provided city, town or province?

Table 1: Provide complete collection date

Table 1: Change “Collecting location” to “Collection location” and “Collecting date” to “Collection date”

Table 7: Size should be in bp not C

Reviewer 2 Report

In the article titled "Full-length SMRT transcriptome sequencing and SSR analysis 2 of Bactrocera dorsalis (Hendel)" authors describe full length transcriptomeby SMRT and describe ssr markers in B.dorsalis. Overall this is a timely study and I would recommend this study to be published. The authors need to extensively edit the manuscript for english language. Figure 3 is low resolution please provide a high resolution image.
